# Novel Methods for Predicting Fluid Responsiveness in Critically Ill Patients—A Narrative Review

**DOI:** 10.3390/diagnostics12020513

**Published:** 2022-02-16

**Authors:** Jan Horejsek, Jan Kunstyr, Pavel Michalek, Michal Porizka

**Affiliations:** 1Department of Anaesthesiology and Intensive Care Medicine, First Faculty of Medicine, Charles University in Prague and General University Hospital in Prague, 12808 Prague, Czech Republic; jan.horejsek@vfn.cz (J.H.); jan.kunstyr@vfn.cz (J.K.); pavel.michalek@vfn.cz (P.M.); 2Department of Anaesthesia, Antrim Area Hospital, Antrim BT41 2RL, UK

**Keywords:** fluid responsiveness, fluid therapy, volume expansion, circulatory shock, hypovolemia, preload, tissue perfusion

## Abstract

In patients with acute circulatory failure, fluid administration represents a first-line therapeutic intervention for improving cardiac output. However, only approximately 50% of patients respond to fluid infusion with a significant increase in cardiac output, defined as fluid responsiveness. Additionally, excessive volume expansion and associated hyperhydration have been shown to increase morbidity and mortality in critically ill patients. Thus, except for cases of obvious hypovolaemia, fluid responsiveness should be routinely tested prior to fluid administration. Static markers of cardiac preload, such as central venous pressure or pulmonary artery wedge pressure, have been shown to be poor predictors of fluid responsiveness despite their widespread use to guide fluid therapy. Dynamic tests including parameters of aortic blood flow or respiratory variability of inferior vena cava diameter provide much higher diagnostic accuracy. Nevertheless, they are also burdened with several significant limitations, reducing the reliability, or even precluding their use in many clinical scenarios. This non-systematic narrative review aims to provide an update on the novel, less employed dynamic tests of fluid responsiveness evaluation in critically ill patients.

## 1. Introduction

Fluid administration represents a first-line therapeutic intervention in patients with acute circulatory failure [1]. The principal goal is to increase stroke volume by increasing cardiac preload, thereby improving cardiac output (CO) and tissue perfusion. The physiologic relationship among cardiac preload, contractility, and stroke volume is described by Frank–Starling law (Figure 1) and creates a scientific basis for the evaluation of fluid responsiveness, which is defined as a significant increase (10–15%) in CO after intravascular volume expansion. Traditionally, such fluid challenge has been achieved by the infusion of 500 mL of either crystalloid or colloid solution over a short period of time. Nevertheless, no standardized technique of fluid challenge has been established so far, as routine practice including type and amount of solution, time of infusion, or resuscitation endpoints varies significantly among authors and institutions [2,3,4]. The paramount problem of the current practice of intensive care medicine is that only approximately 50% of critically ill patients are “fluid responders” [5]. Furthermore, based on a large international multicentre study, fluid responsiveness is not predicted routinely and its evaluation techniques vary widely across institutions [6]. That means that a large proportion of patients receive a significant amount of unnecessary fluid, which subsequently leads to hyperhydration with the development of interstitial and intracellular tissue oedema [7]. Such fluid overload has been shown to adversely influence morbidity and survival in a wide range of critically ill patients due to compromised organ function [8,9,10,11,12]. For these reasons, it seems rational that fluid therapy should be tailored to individual patients’ needs with regard to their actual volume status in the course of the disease, although some authors, as well as guidelines, make a case for fixed volumes of infusion [13,14].

The decision to administer fluids has been traditionally based on the clinical signs of hypovolaemia, such as hypotension, oliguria, increased serum lactate, and static parameters of preload including central venous pressure or pulmonary artery wedge pressure [15]. Unfortunately, despite its widespread use, none of these parameters are reliable in predicting fluid responsiveness alone [13,16,17,18,19] or in combination [20]. Thus, such low predictive value of these traditional markers has led to the adoption of dynamic tests, which rely on the induction of changes in cardiac preload, eliciting a response to such fluid challenge without administering fluid [21]. Some dynamic techniques are based on the heart–lung interactions during positive pressure ventilation, such as pulse pressure variation (PPV), stroke volume variation (SVV), or the collapsibility of inferior and superior vena cava. In these methods, cyclic changes in intrathoracic pressure during positive pressure ventilation result in changes in the preload of both cardiac ventricles, acting as an internal fluid challenge [22]. The passive leg raising (PLR) manoeuvre also creates an internal fluid challenge by increasing venous return temporarily [23]. The standard fluid challenge can be substituted with the mini-fluid challenge, which reduces the volume of fluid inadvertently administered to non-responsive patients [24].

Despite a very high degree of accuracy in predicting fluid responsiveness, especially in the case of PLR and PPV, all these methods have several significant limitations, making them impractical or even impossible to use reliably in certain clinical scenarios [25]. For instance, spontaneous breathing activity, small tidal volume ventilation, and irregular cardiac rhythm represent major limitations of PPV and SVV [26], with the proportion of eligible patients in intensive care reported as low as 2% for PPV [27]. On the other hand, PLR is limited only by the use of compression stockings [28] and intraabdominal hypertension [29]; however, the volume of blood transferred to the central compartment (approximately 300 mL) is lower than the conventional volume challenge and requires sufficiently precise monitoring methods of CO [30]. These include either more or less invasive methods such as pulse contour analysis or pulmonary artery catheter thermodilution and non-invasive echocardiography requiring a trained operator, who may not always be available. Thus, to overcome these limitations and drawbacks, novel methods of predicting fluid responsiveness have been introduced recently including respiratory occlusion tests, jugular vein collapsibility, or arterial doppler variation. The aim of this review is to analyse the existing evidence and provide up-to-date information on novel methods of predicting fluid responsiveness in critically ill patients.

## 2. Materials and Methods

Our aim was to identify all manuscripts relevant to the methods of predicting fluid responsiveness described in this article. For the purpose of this non-systematic narrative review, we performed a comprehensive electronic and manual search of the following databases: PubMed, Scopus, Web of Science, and Google Scholar, within the time frame of January 1990 to September 2021. Systematic reviews, meta-analyses, narrative reviews, randomized controlled trials, prospective and retrospective cohort studies, case reports, and letters to the editor were all included. Additional manuscripts were identified from the reference lists of screened articles. No language restriction was applied, although the information for non-English manuscripts was retrieved from the abstracts only. Only published or accepted manuscripts accessible online ahead of print were processed. Animal, cadaver, and manikin studies were excluded. The terms used included “fluid responsiveness”, “hypovolemia”, “preload”, “volume expansion”, “expiratory occlusion”, “inspiratory occlusion”, “arterial Doppler”, “vein variability”, “extrasystole”. J.H. and M.P. performed the initial search independently, while P.M. and J.K. reviewed their results and provided additional relevant studies.

## 3. End-Expiratory and End-Inspiratory Occlusion

The end-expiratory occlusion test (EEOT) was first described in the study by Monnet et al. in 2009 [31] as one of the dynamic parameters of fluid responsiveness based on heart–lung interactions. Periodic insufflation during positive pressure ventilation increases intrathoracic pressure, which results in decreased venous return and preload of the right and subsequently of the left ventricle. Interrupting respiration at the level of positive end-expiratory pressure (PEEP) with a decrease in intrathoracic pressure should therefore increase the preload of both ventricles, resulting in an “internal” fluid challenge. In a fluid-responsive patient, such an increase in preload translates into a significant elevation of CO. EEOT has been successfully tested in mixed ICU and surgical patient populations [31,32,33,34,35,36,37].

In mechanically ventilated patients, EEOT is performed by interrupting the respiratory cycle at end expiration. This can be achieved, for instance, by using the expiratory hold function on the ventilator, routinely employed to measure intrinsic PEEP. Typically, an end expiratory hold lasting 15 s has been described in most of the studies [31,32,33,34,37,38,39,40]; however, durations of 12 [41] or 30 s [35,36,42] have also been reported. A recent meta-analysis showed no diagnostic benefit of expiratory holds lasting more than 15 s [43]. To reliably assess the hemodynamic effect of EEOT, CO or its surrogates have been monitored [44]. Ideally, the measurement method of choice should provide a sufficiently rapid and precise calculation of CO, as the magnitude of changes induced by EEOT is presumed to occur in the last 5 s of the end-expiratory occlusion and wane entirely within 1 min after the test completion [31]. Thus, calibrated pulse-contour analysis has been applied in the majority of the studies [31,33,34,37,40,45]; however, uncalibrated pulse-contour analysis [35,46], echocardiography [32,41], oesophageal Doppler [39,47], plethysmography-derived estimations of CO [38], and even pulse pressure changes as a CO surrogate [31] have also been used with acceptable precision. Importantly, the CO changes induced by EEOT are of relatively low amplitude, which partially explains why monitoring changes in end-tidal CO_2_ does not reliably predict fluid responsiveness during EEOT [48]. In a study with septic ICU patients, a CO increase of only 5% after EEOT measured by calibrated pulse-contour analysis predicted fluid responsiveness with excellent diagnostic performance (sensitivity of 91%, specificity of 100%, and ROC AUC of 0.972) [31]. Similar results were obtained in further studies in mixed medical and surgical populations of ICU patients with both calibrated [32,33,34,37] and non-calibrated pulse-contour analysis [35]. Nevertheless, such low-amplitude changes in CO during EEOT represent a major limitation if ultrasound techniques of assessing CO are used. When oesophageal Doppler was used, a mild increase in CO of 2.3% [47] or 4% [39] predicted fluid responsiveness. The measurement of velocity-time integral (VTI) of the left ventricular outflow tract (LVOT) was examined in one study with a CO change threshold of 5% [36] using transoesophageal echocardiography (TOE) and in two studies with a threshold of 5–9% using transthoracic echocardiography (TTE) [32,41]. However, these values are lower than the least significant change, i.e., the minimal change between two measurements that can be considered significant. These were calculated to be 11% for LVOT VTI assessed by TTE [49] and 7% for oesophageal Doppler [39]. Thus, the end-inspiratory occlusion test (EIOT) was introduced as a complement to EEOT to improve the diagnostic accuracy of the test by increasing the magnitude of CO changes. The principle of EIOT is similar to EEOT when insufflations are interrupted in the inspiratory phase, resulting in a decrease in preload. By adding the absolute values of changes induced by EEOT and EIOT separated by one minute, Jozwiak et al. were able to increase the fluid responsiveness detection threshold from 4% to 11% of CO change monitored by pulse contour analysis [32]. The same effect was also achieved for LVOT VTI assessed by TTE [32] and oesophageal Doppler [39].

The major limitation of EEOT is that it can only be performed in mechanically ventilated patients without any spontaneous breathing activity that would preclude a 15-s expiratory hold. PEEP levels between 5 and 14 cm H_2_O do not seem to influence the diagnostic accuracy of the test [34]. The tidal volumes of ≈7 mL/kg have been successfully tested in numerous studies [31,33,34,35,40,41]. On the other hand, only limited data exist on the diagnostic performance of these tests in patients with low tidal-volume ventilation [37,46] and when prone positioning is used [45]. The main advantage of EEOT lies in its rapid execution and utility even in the operating theatre, where other markers of fluid responsiveness might be impractical.

## 4. Internal Jugular Vein Distensibility

Another potentially useful dynamic parameter of fluid responsiveness is the respiratory variation of internal jugular vein (IJV) diameter, called the IJV distensibility/variability in mechanically ventilated patients, IJV collapsibility in spontaneously breathing patients, or jugular index [50,51,52], all evaluated by vascular ultrasound. Like superior or inferior vena cava distensibility, the test is based on cyclical changes in venous return to the heart induced by intra-thoracic pressure variation during mechanical ventilation or spontaneous breathing (Figure 2A,B).

The recommended technique requires a patient to be placed either in the supine [51,52] or 30° semi-recumbent position [50,53]. A linear ultrasound probe should be positioned perpendicular to the skin in the transverse plane of the neck at the level of the cricoid cartilage. The IJV is then identified using colour Doppler imaging and by direct external compression. The IJV diameter is evaluated using ultrasound M-mode, and the highest (D_max_) and lowest diameter (D_min_) values during one respiratory cycle are recorded. The measurement of anterior–posterior (AP) IJV diameter has been most commonly used [50,53,54,55]; nevertheless, transverse diameter yielded similar results as AP measurements in one experimental study [51]. It is of paramount importance to avoid any venous compression with the ultrasound probe during measurement. The formula for IJV distensibility (IJVD) varies between the studies; however, it is most frequently calculated as: (1)$$\mathrm{IJVD}\left(\%\right)=\frac{D_{\max}-D_{\min}}{D_{\max}}\times 100$$

On the other hand, if IJV variability (*IJVV*) is used, the calculation formula is modified to: (2)$$IJVV\left(\%\right)=\frac{D_{\max}-D_{\min}}{(D_{\max}+D_{\min})/2}\times 100$$

In a cohort of 50 mechanically ventilated patients in septic shock, IJVD provided superior results with the threshold of 18% predicting an increase in CO ≥ 15% induced by a fluid challenge (sensitivity of 80%, specificity of 95%, and ROC AUC of 0.915) [50]. In another study [53] with 70 mechanically ventilated cardiac surgical patients, *IJVV* > 12.99% predicted an increase in SV ≥ 15% induced by a fluid challenge with similar accuracy. There are only limited data for spontaneously breathing patients. In the study conducted by Haliloglu et al. [54], IJV collapsibility was evaluated in 44 spontaneously breathing patients in sepsis. IJV collapsibility ≥ 36% predicted an increase in CO ≥ 15% after a passive leg raising (PLR) manoeuvre with a moderate sensitivity of 78% and a specificity of 85%, and an ROC AUC of 0.872. In another experimental study, there was a significant difference in IJV collapsibility in healthy volunteers before and after donating 450 mL of blood [51]; however, no monitoring of CO was performed. In a study with patients receiving pressure support ventilation, changes in IJV collapsibility predicted fluid responsiveness reliably (sensitivity of 83%, specificity of 94%, and ROC AUC of 0.88), though interestingly only when measurements were conducted in the right internal jugular vein [56].

The main advantage of IJVD compared to the more established distensibility of inferior vena cava lies in its lower technical difficulty and easier application, especially in patients with obesity, ascites, or intra-abdominal hypertension [50,57]. Furthermore, the evaluation of distensibility of superior vena cava necessitates the introduction of an oesophageal echocardiographic probe and a trained operator, whereas only a linear probe and basic ultrasound expertise are required for IJV distensibility measurement [50]. The major limitations of IJVD are presumed to be low lung compliance, cardiac arrhythmias, or jugular vein thrombosis. Low tidal volume ventilation is also likely to impair the diagnostic accuracy of the test. Guarracino et al. [50] used tidal volumes of 6–8 mL/kg with acceptable results, but further research is needed to confirm these findings. Nonetheless, IJVD appears to be a promising parameter of fluid responsiveness, especially advantageous for its rapid, non-invasive evaluation without the requirement of advanced ultrasound skills.

## 5. Hepatic Venous Flow

The respiratory variation of the inferior vena cava (IVC) diameter has been widely used as a reliable predictor of fluid responsiveness, although its diagnostic reliability has been questioned recently, especially in the cohort of spontaneously ventilating patients [58,59]. It has been hypothesized that measuring the flow instead of diameter in the vena cava might be a better indicator of preload, but since its angular appearance in the subcostal ultrasound views precludes accurate flow Doppler measurements, hepatic venous flow (HVF) was used as a surrogate [60]. The ultrasound Doppler analysis waveform of a hepatic vein physiologically consists of four waves (two antegrade waves S and D, one retrograde wave A, and one transitional wave V) (Figure 3) [61].

Nevertheless, the hepatic vein spectral waveform may become atypical under many pathological conditions, such as a change in the flow direction (hepatic vein occlusion), irregularity (arrhythmias), mono-/biphasic shape (hepatic infiltration, cirrhosis, intra-abdominal hypertension), or D-wave dominance (tricuspid regurgitation, right heart failure) [61] (Figure 4).

In a prospective study with 44 mechanically ventilated patients in septic shock, HVF was evaluated from subcostal echocardiographic views by first scanning the IVC and then identifying the middle hepatic vein (MHV) [60]. Pulse Doppler was used to assess the blood flow characteristic in the MHV, and the peak flow velocities of each individual wave were recorded. The change in MHV D-wave velocity (ΔMHV D) after volume expansion (VE) was calculated as: (3)$$\Delta \mathrm{MHV}\;D\left(\%\right)=\frac{\left(\mathrm{MHV}\;D_{\mathrm{after}\;\mathrm{VE}}-\mathrm{MHV}\;D_{baseline}\right)}{\mathrm{MHV}\;D_{baseline}}\times 100$$

The authors concluded that none of the evaluated parameters were able to predict fluid responsiveness. However, an increase in ΔMHV D ≥ 21% during volume expansion was associated with a lack of fluid responsiveness with a sensitivity of 100%, specificity of 71%, and high ROC AUC of 0.918. Furthermore, MHV S-wave velocity significantly correlated with CO, as it increased after fluid challenge in the fluid responders but did not change in the non-responder group. Thus, HVF evaluation seems to offer an indicator for halting ongoing fluid administration and may serve as a surrogate to assess cardiac output changes. The main advantage of evaluating HVF is its lower technical difficulty and better availability in patients in whom obtaining standard TTE views might prove impossible, especially those after cardiac surgery. The major limitations represent the various concurrent pathologies affecting the hepatic blood flow pattern as described above. Further studies are warranted to validate the clinical utility of HVF assessment and its role in the fluid management of critically ill patients.

## 6. Arterial Doppler Monitoring

Arterial Doppler measurements may provide valuable information about patients’ circulatory status and CO while being entirely non-invasive. Historically, oesophageal Doppler has been widely used to continuously monitor various haemodynamic variables including CO, which correlates closely with CO measured by a pulmonary artery catheter using the thermodilution method [62]. Oesophageal Doppler can also be used to measure corrected systolic flow time (FTc), which was found to correlate with intravascular volume [63]. FTc is the systolic portion of the cardiac cycle with correction for the heart rate (division by the square root of cycle time—Bazett’s formula).
(4)$$\mathrm{FTc}=\frac{\mathrm{raw}\;\mathrm{flow}\;\mathrm{time}}{\sqrt{\mathrm{cycle}\;\mathrm{time}}}$$

As FTc is considered to be a static parameter of preload, it was mostly used in conjunction with the passive leg raising (PLR) manoeuvre. In other studies, arterial blood flow (BF) was measured instead, obtained by using the standard formula: (5)$$\mathrm{BF}=\pi \times \frac{{\left(\mathrm{arterial}\;\mathrm{diameter}\right)}^{2}}{4}\times \mathrm{velocity}\;\mathrm{time}\;\mathrm{integral}\left(\mathrm{VTI}\right)\times \mathrm{heart}\;\mathrm{rate}$$

Nevertheless, such calculation of aortic flow requires the use of an oesophageal Doppler that may not be available in all patient settings. To overcome this limitation, the investigation of more easily accessible arteries has been explored, including the carotid, brachial, femoral, and splenic arteries. Most available data on arterial ultrasound Doppler monitoring focused on the carotid artery.

### 6.1. Carotid Artery

In a study with 34 haemodynamically unstable patients in septic shock, Marik et al. [3] demonstrated that a 20% increase in carotid blood flow following the PLR manoeuvre predicted fluid responsiveness with high accuracy (sensitivity of 94%, specificity of 86%). On the other hand, brachial artery blood flow simultaneously increased to a much lesser extent, providing lower diagnostic accuracy of the test. The authors speculated that such phenomenon could be explained by the physiologic redistribution of blood flow in the distributive type of circulatory shock. Furthermore, Blehar et al. hypothesized that clinically dehydrated patients would have an increase in carotid corrected flow time (CFTc, Figure 5) following volume expansion. In their observational study, 56 patients who were considered hypovolemic based solely on medical history and clinical examination were given a fluid bolus. CFTc increased from 299 to 340 ms with MAP and HR remaining unchanged [64]. Unfortunately, no other haemodynamic parameters were measured in that study. Similar increases in CFTc were observed in a population of blood donors [65], end-stage renal failure patients on dialysis [66,67], and healthy volunteers after prolonged fasting [68], but every time without recording any haemodynamic parameters except for MAP and HR, thus not correlating changes in CFTc with CO. In another study, Barjaktarevic et al. [69] enrolled 77 patients with undifferentiated shock, monitoring cardiac output by a system based on bioreactance. The CFTc change threshold value of 7 ms was identified as the best predictor of fluid responsiveness with a high degree of diagnostic accuracy (ROC AUC 0.88). On the other hand, Judson et al. observed an increase in MAP after administering a fluid challenge to patients in septic shock without a corresponding change in CFTc [70].

### 6.2. Brachial Artery

Respiratory variability of brachial artery peak velocity (ΔVpeak_brach_) was investigated as a predictor of fluid responsiveness in 38 mechanically ventilated patients in a study by Garcia et al. [71]. ΔVpeak_brach_ was measured prior to scheduled volume expansion and calculated using the following formula:(6)$${\Delta \mathrm{Vpeak}}_{\mathrm{brach}}\left(\%\right)=100\times \left({\mathrm{Vpeak}}_{\max}-{\mathrm{Vpeak}}_{\min}\right)\;/\;\frac{{\mathrm{Vpeak}}_{\max}+{\mathrm{Vpeak}}_{\min}}{2}$$

ΔVpeak_brach_ of more than 10% predicted fluid responsiveness with high accuracy (ROC AUC 0.88) comparable to PPV. This was confirmed by another work by Brennan et al. [72] who found a strong correlation (r = 0.84) between these two parameters.

### 6.3. Femoral Artery

Peak velocity of femoral artery flow and femoral velocity time integral variation have been evaluated in two studies in ICU patients with sepsis. Préau et al. [73] used a variety of parameters to predict fluid responsiveness including the changes in femoral artery flow velocity (ΔVpeak*_femoral_*). Changes in CO were evaluated by measuring stroke volume with TTE. ΔVpeak*_femoral_* was calculated for the PLR manoeuvre and fluid challenge as:(7)$${\Delta \mathrm{Vpeak}}_{femoral}\left(\%\right)=100\times \frac{\left({\mathrm{Vpeak}}_{after}-{\mathrm{Vpeak}}_{baseline}\right)}{{\mathrm{Vpeak}}_{baseline}}$$

A PLR-induced ΔVpeak*_femoral_* ≥ 8% predicted fluid responsiveness with high accuracy (ROC AUC 0.93) and correlated with increased CO after fluid challenge; this parameter performed similarly to PPV measured by radial artery catheter. These results were subsequently replicated in the study of Luzi et al. in a mixed population of critically ill patients [74].

### 6.4. Splenic Artery

Splenic Doppler resistive index (SDRI) has also been proposed as a non-invasive parameter to detect hypovolemia. The technique is based on the ultrasound Doppler measurements obtained by placing the sample gate into the main branches of the splenic artery, about 1 cm past the splenic hilum. SDRI is calculated using the following formula: (8)$$\mathrm{SDRI}=\frac{S-D}{S}$$
where *S* stands for the peak systolic velocity and *D* for the end-diastolic velocity. In a study by Corradi et al. [75], SDRI was evaluated in 49 haemodynamically stable patients with polytrauma on admission. Significantly higher SDRI was observed in patients with occult bleeding, developing haemorrhagic shock within 24 h after admission, compared to stable patients without bleeding episodes (SDRI of 0.71 vs. 0.6, respectively). After volume resuscitation, the SDRI of bleeding patients was similar to the values in the non-bleeding group. In another study with cardiac surgical patients on mechanical ventilation, Brusasco et al. [76] showed that a decrease in SDRI < 4% after standard volume expansion excluded fluid responsiveness with 100% of sensitivity and specificity; on the other hand, a decrease ≥ 9% reflected fluid responsiveness (sensitivity 63%, specificity 100%, ROC AUC 0.88).

In conclusion, the main advantages of evaluating arterial Doppler include the complete non-invasiveness and simplicity of the measurement, as all the arteries besides the splenic artery lie superficially and are easily examined by ultrasound, making this method especially handy in the emergency department. Regarding limitations, most published studies have focused on patients outside of the intensive care setting and data for critically ill patients are limited. Furthermore, in a large portion of available studies, CO was not monitored directly, and its eventual changes were only assumed as a consequence of intravascular volume reduction or upon clinical presentation [64,65,66,67,68]. Lastly, despite the aforementioned technical simplicity of this technique, there are several studies indicating low reproducibility between sonographers [77] and insufficient reliability in patients undergoing the PLR manoeuvre [78]. Generally, the current data available for brachial, femoral, and splenic artery flow evaluation are insufficient for implementation into the routine practice of fluid responsiveness evaluation. 

## 7. Extrasystoles

Most of the methods currently used for the evaluation of fluid responsiveness require advanced haemodynamic monitoring, which may not be readily available to all critically ill patients. Thus, to overcome such technical limitations, Vistisen et al. proposed that cardiac extrasystoles may be regarded as a preload modifying mechanism and their hemodynamic effect could be monitored using only standard electrocardiographic (ECG) and invasive arterial blood pressure (BP) measurements [79]. This hypothesis is based on the fact that the prematurely occurring extrasystole, or ectopic beat, is associated with decreased ventricular filling. On the other hand, the post-ectopic beat is associated with increased preload due to the preceding compensatory pause and may result in an increase in stroke volume in the fluid responsive patient [80]. Thus, the monitored variables include post-ectopic systolic blood pressure (SBP) and the pre-ejection period (PEP), defined as the time interval between the R-wave on ECG and the onset of the systolic upstroke of arterial BP, both compared with the median values of ten preceding sinus beats. In cases where multiple eligible extrasystoles were detected, these variables were also averaged into a single value. In another study by Vistisen et al. [80], ECG and arterial BP waveform within a period of 30 min prior to a scheduled fluid challenge were analysed in a population of 41 critically ill patients. Fluid responsiveness was defined as a 10% increase in SV evaluated by non-invasive cardiac output monitoring. The extrasystoles were detected in 63% of patients. A post-ectopic increase in SBP of 5% and an increase in PEP of 7.5 ms predicted fluid responsiveness with only moderate accuracy (ROC AUCs of 0.79 and 0.74, respectively). Similar results were also obtained in a cohort of surgical patients in a study of the same investigator group [81]. On the other hand, in the study conducted intraoperatively in 61 cardiac surgical patients, analysis performed twice during the procedure identified only 41% and 46% of eligible extrasystoles. In this case, the post-ectopic beat variables, including SBP and PEP, predicted fluid responsiveness less reliably (with ROC AUC ranging from 0.5 to 0.69) [82].

The main advantage of using extrasystoles as a predictor of fluid responsiveness lies in its modest technical requirements, where only ECG and arterial BP waveform analysis are necessary. The method should be applicable and feasible in a large number of ICU patients. There are several significant limitations, however. First, the clinician is compelled to wait for an extrasystole to occur and unless the monitoring device used offers sufficient post-hoc analysis, an observer would be required to stand at the bedside, constantly checking the monitor. Thus, wider availability of automated ECG analysis software is required to make its application more convenient. Second, only patients with sinus rhythm have been evaluated so far. Third, there are no data on the effects of extrasystole timing in relation to the respiratory cycle [80]. Lastly, all the studies that have been conducted to date include relatively small numbers of patients and all were authored by one group of researchers with publication in only one journal.

## 8. Limitations

The major limitation of the presented non-systematic review lies in the fact that it includes only small, non-randomized and single-centre studies, even including some lacking any reproducibility. Additionally, there has never been a direct comparison of the presented methods between them alone or to more established and widely used techniques of fluid responsiveness evaluation. Therefore, it is not clear whether any of the methods are superior to any other or which one should be used preferentially. Finally, the narrative nature of the review may lead to a selection and interpretation bias as it does not follow objective evidence-based criteria of study inclusion and evaluation used in systematic reviews.

## 9. Conclusions

Adequate fluid therapy belongs to some of the most challenging yet also most essential skills of all intensive care practitioners as it has a paramount effect on a patient’s final clinical outcome. The evaluation of fluid responsiveness offers a means of differentiating patients in whom fluid administration will provide an increase in cardiac output; nevertheless, choosing the appropriate test with respect to its limitations, the clinical situation, as well as available equipment, may prove difficult. The presented novel methods of evaluating fluid responsiveness also have limitations and drawbacks; however, if used in a complementary fashion to the already-established techniques, they can provide additional valuable information to decide whether the patient requires fluid administration. In addition, the non-invasiveness of these methods represents a major advantage in highly vulnerable critically ill patients prone to iatrogenic complications. Most importantly, it is necessary to realize that fluid responsiveness is not per se an indication for volume administration. Fluid therapy should always be indicated based on clinical signs of hemodynamic instability and peripheral tissue hypoperfusion with regard to the individualized risk/benefit ratio.

## Figures and Tables

**Figure 1 diagnostics-12-00513-f001:**
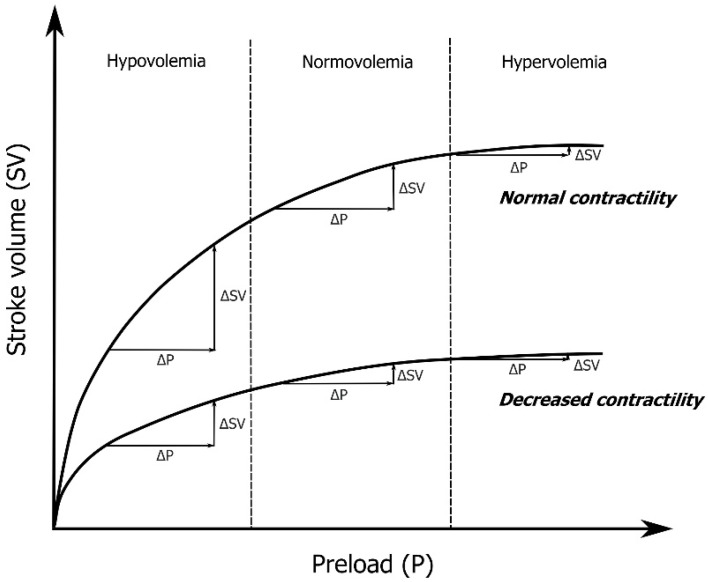
The Frank–Starling mechanism representing the relationship of myocardial contractility (stroke volume) and cardiac preload.

**Figure 2 diagnostics-12-00513-f002:**
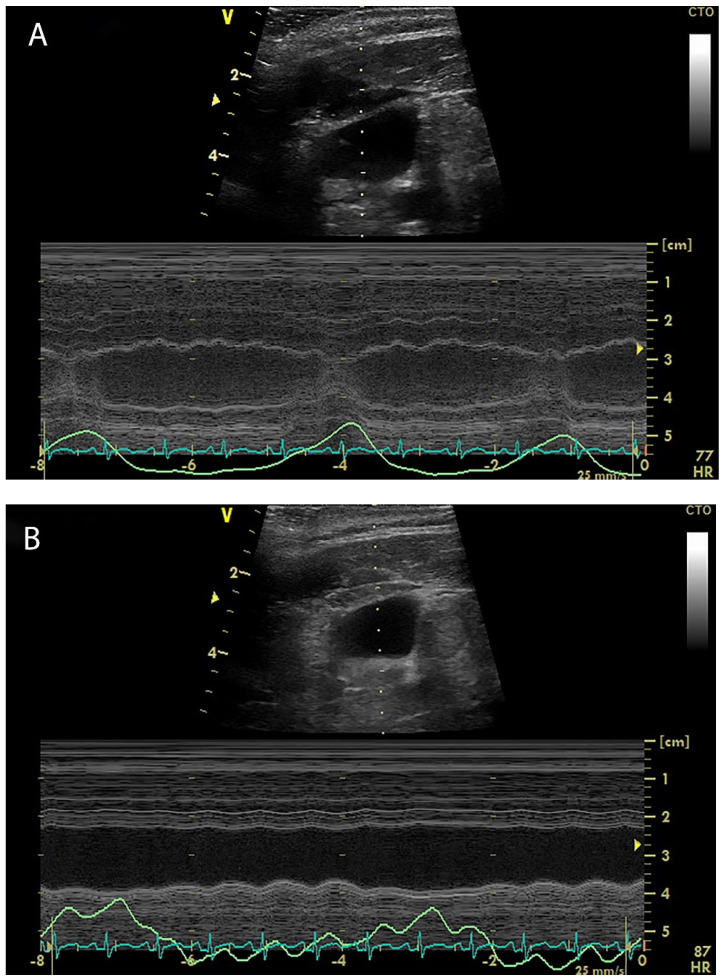
M-mode view of the antero-posterior diameter of the internal jugular vein shows differences in variability in a fluid responsive (**A**) and non-responsive (**B**) mechanically ventilated patient.

**Figure 3 diagnostics-12-00513-f003:**
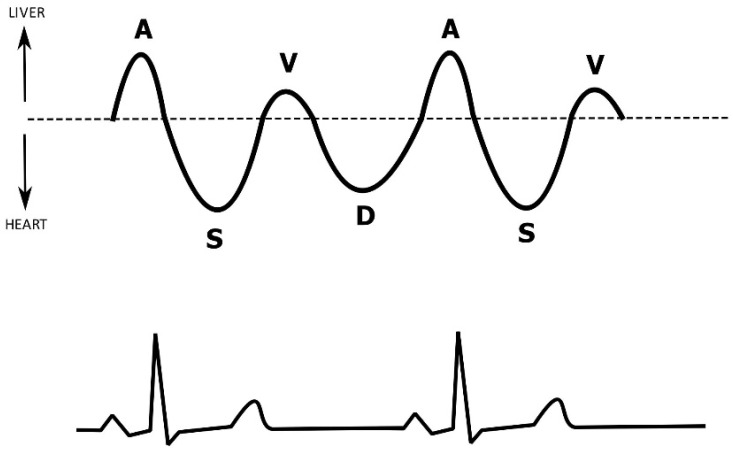
A diagram of the normal spectral Doppler waveform in the middle hepatic vein (MHV) correlated with a concurrent ECG tracing. During the cardiac cycle, atrial systole results in a retrograde flow of blood toward the liver, producing the A wave. The S wave is seen during ventricular systole, when antegrade blood flow in the MHV is produced as the tricuspid valve moves toward the cardiac apex. Afterwards, the tricuspid valve returns to its original position and the retrograde V wave is seen. Ventricular diastole is associated with passive blood inflow from the atria, producing the D wave.

**Figure 4 diagnostics-12-00513-f004:**
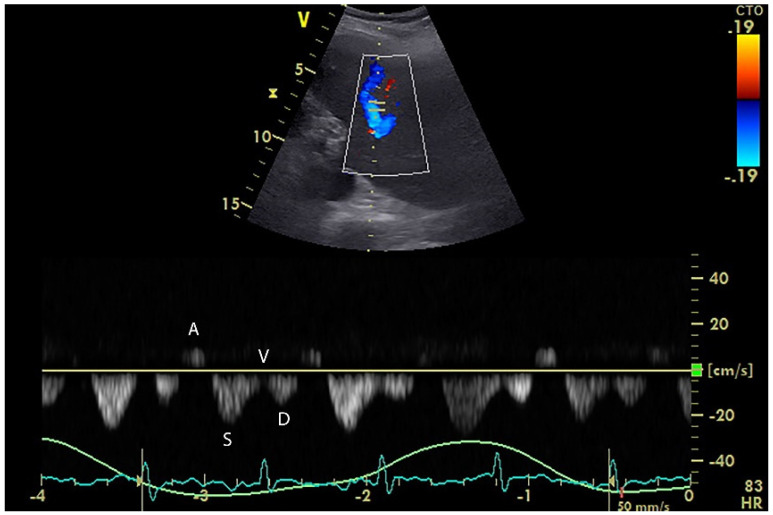
Spectral Doppler waveform in the middle hepatic vein (MHV) of a ventilated patient.

**Figure 5 diagnostics-12-00513-f005:**
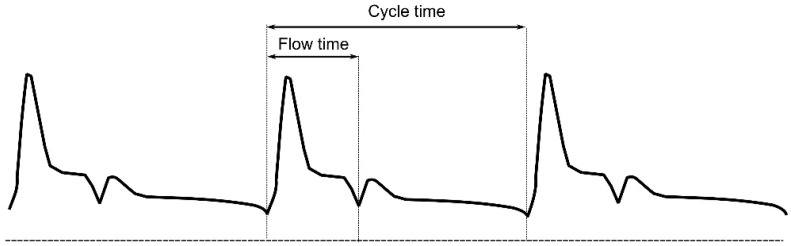
A diagram of the normal spectral Doppler waveform in the common carotid artery showing the measurement of the carotid flow time (CFT). The flow time is measured from the beginning of the carotid upstroke to the central portion of the dicrotic notch. The corrected carotid artery flow time (CFTc) is calculated using the Bazett’s formula explained above.

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
