# Peer review of "Novel Methods for Predicting Fluid Responsiveness in Critically Ill Patients—A Narrative Review"

_diagnostics, 2022, doi:10.3390/diagnostics12020513_

Round 1

Reviewer 1 Report

A brief but comprehensive review of the latest methods for identifying critically ill patients who are fluid responders. Congratulations to the Authors. I do not see the need for corrections.

Author Response

Answer: A professional language revision by native English speaker (MDPI author services) was performed.

Reviewer 2 Report

Non systematic review of novel methods for predicting fluid responsivness in critically ill patients. Well connected and written. 

Specific comments

  • Use British English throughout text
  • Page 2, lines 69-70: Muscle relaxation and controlled ventilation are required for PLR. The sentence should be adapted
  • Methods: in the title, abstract and section Materials and Methods, it should be mentioned that this is a non-systematic review
  • 6. Extrasystoles is a parameter only assessed by one group (as mentioned) and published in one journal only. Suggest to mention it as the last parameter before Conclusions
  • A Limitations section should be added before the Conclusions

Author Response

Specific comments

  • Use British English throughout text

Answer: Professional language revision was performed and British English is used throughout the text (MDPI author services).

  • Page 2, lines 69-70: Muscle relaxation and controlled ventilation are required for PLR. The sentence should be adapted

Answer: PLR maneuver has been succesfully used in many clincial settings even in patients with spontaneous ventilation (Cherpanath TG, Hirsch A, Geerts BF, et al. Predicting Fluid Responsiveness by Passive Leg Raising: A Systematic Review and Meta-Analysis of 23 Clinical Trials. Crit Care Med. 2016;44(5):981-991. doi:10.1097/CCM.0000000000001556), therefore we would prefer to make no corrections in this sentence.

  • Methods: in the title, abstract and section Materials and Methods, it should be mentioned that this is a non-systematic review

Answer: We emphasised the fact that this review is non-systematic and narrative in the title, abstract and Materials and Methods as reviewer proposed.

  • 6. Extrasystoles is a parameter only assessed by one group (as mentioned) and published in one journal only. Suggest to mention it as the last parameter before Conclusions

Answer: A note that the studies on extrasystoles was published in only one journal was added in the text (page 11, line 387-389) and we transferd the whole section before Conclusions.

  • A Limitations section should be added before the Conclusions

Answer: A Limitations section was added before Conclusions (page 11, line 391-400).

Reviewer 3 Report

Thank you for the opportunity of revising the manuscript entitled “Novel methods for predicting fluid responsiveness in critically ill patients” by Horejsek et al.

In this review the authors provide an overview on some techniques for detecting preload responsiveness in intensive care unit patients. Even though the topic is of great interest in critical care medicine, the manuscript has too many flaws that in my opinion do not make it suitable for publication in the current form. In particular, the manuscript is often not fluent and not coherent. Also, it fails to provide new information on the subject (more clearly, what does this review add to the other ones?)

Please refer to comments below.

The quality of the English language is poor (very poor in some sections). I strongly suggest revision by a native English speaker.

The introduction presents many imprecise points:

  • Line 32: it should be better detailed how to perform a fluid challenge (type of fluid, time and amount – are all the major authors unanimous?). Please refer to Vincent et al Fluid challenge revisited Crit Care Med
  • Line 34: the survey you refer to was published in 2015, so not so recent.
  • Figure 1: the graph you present is not the relationship between myocardial contractility and cardiac preload, but rather the relationship between stroke volume and cardiac preload. As a matter of fact, there is not A Frank-Starling curve, but rather a family of curves depending on the contractility status

Materials and methods: the authors present this section as if the manuscript were a systematic review. However, no string search, no flow chart of study inclusion, no results of the search are provided. This is a major flaw.

The end-expiratory occlusion test is now thirteen years old, which is only three years older than the PLR as described by Monnet and Teboul. Apart from that, the section is quite no more than a synthesis of the results and the discussion of the recent meta-analysis cited by the authors.

Minor comments

Line 145-146: the study of Jozwiak et al (ref 46) focused on TTE, whereas the one of Depret et al focused on oesophageal doppler (ref 38).

Line 201: a reference should be added to the sentence regarding the advantages of IJV distensibility measurement.

Lines 208-201: same comment as the previous one.

Round 2

Reviewer 3 Report

Unfortunately I still do not believe that the current version has sufficiently improved to recommend publication.

Author Response

We answered all the questions and made all the required corrections including English revision by native speaker (MDPI author services) from the first round review as requested by the reviewer. As no specific requirements or questions were raised in the 2nd round, we do not know how to react to this. We fully stand behind what we wrote and the total rewriting the of the whole article is not acceptable for us unless specific queries are made.